# A Method for Integration of Preferences to a Multi-Objective Evolutionary Algorithm Using Ordinal Multi-Criteria Classification

Alejandro Castellanos-Alvarez [1], Laura Cruz-Reyes [1], Eduardo Fernandez [2], Nelson Rangel-Valdez [3], Claudia Gómez-Santillán [1,*], Hector Fraire [1] and José Alfredo Brambila-Hernández [1]

1   Graduate Program Division, Tecnológico Nacional de México, Instituto Tecnológico de Ciudad Madero, Cd. Madero 89440, Mexico; alex810_castellanos@hotmail.com (A.C.-A.); lauracruzreyes@itcm.edu.mx (L.C.-R.); automatas2002@yahoo.com.mx (H.F.); alfredo.brambila@outlook.com (J.A.B.-H.)
2   Research and Postgraduate Directorate, Universidad Autonoma de Coahuila, Saltillo 26200, Mexico; eddyf171051@gmail.com
3   CONACyT Research Fellow at Graduate Program Division, Tecnológico Nacional de México, Instituto Tecnológico de Ciudad Madero, Cd. Madero 89440, Mexico; nelson.rangel@itcm.edu.mx
*   Correspondence: claudia.gomez@itcm.edu.mx

**Abstract:** Most real-world problems require the optimization of multiple objective functions simultaneously, which can conflict with each other. The environment of these problems usually involves imprecise information derived from inaccurate measurements or the variability in decision-makers' (DMs') judgments and beliefs, which can lead to unsatisfactory solutions. The imperfect knowledge can be present either in objective functions, restrictions, or decision-maker's preferences. These optimization problems have been solved using various techniques such as multi-objective evolutionary algorithms (MOEAs). This paper proposes a new MOEA called NSGA-III-P (non-nominated sorting genetic algorithm III with preferences). The main characteristic of NSGA-III-P is an ordinal multi-criteria classification method for preference integration to guide the algorithm to the region of interest given by the decision-maker's preferences. Besides, the use of interval analysis allows the expression of preferences with imprecision. The experiments contrasted several versions of the proposed method with the original NSGA-III to analyze different selective pressure induced by the DM's preferences. In these experiments, the algorithms solved three-objectives instances of the DTLZ problem. The obtained results showed a better approximation to the region of interest for a DM when its preferences are considered.

**Keywords:** incorporation of preferences; multi-criteria classification; decision-making process; multi-objective evolutionary optimization; outranking relationships

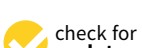

## 1. Introduction

Many industrial domains are concerned with multi-objective optimization problems (MOPs), which in general have conflicting objectives to handle [1]. To solve optimally, a MOPs is to find a set of solutions defined as Pareto optimal solutions. They represent the best compromise between the conflicting objectives. A promising alternative is solving MOPs with metaheuristics, like multi-objective evolutionary algorithms (MOEAs); they obtain an approximation of the Pareto optimal set. This approach solves the problem partially. The decision-maker (DM) has to choose the best compromise solution, which satisfies his preferences, from the set of solutions obtained (non-dominated by each other). For practical reasons, the DM needs to choose one solution to implement it.

MOEAs face various problems when dealing with many objectives—exponential growth in the number of non-dominated solutions and high computational cost to maintain population diversity [2–4], among others. In addition to the previous problems, decision-making becomes difficult when the number of objectives increases.

One way to reduce the DM's cognitive effort is to consider the preferences to guide the MOEA to the region of interest (ROI). Incorporating DM's preferences requires considering non-trivial aspects—defining the DM's preferences, determining the ROI and determining the relevance of a solution [5]. The preferences incorporation methods have used the following representation structures [6,7]—weights, ranking of solutions, ranking of objective functions, reference point, trade-offs between objective functions, desirability thresholds, outranking relations. This paper incorporates preferences using outranking relations.

In many real-world situations, the MOPs environment implicates imprecise information derived from inaccurate measurements or the variability in DMs' judgments and beliefs. Not considering these imprecisions can lead to unsatisfactory solutions and, in consequence, to a poor choice between the existing alternatives due to imperfect knowledge of the problem [8]. Imprecise information may be present in different MOP components; for example, it can be either in objective functions, restrictions, or a decision-maker's preferences. Obtaining the preferential model parameters is a difficult task that increases with the objective number, only possible when the handle of imprecision is allowed [9]. The simplest approach to handling imprecise information is to estimate this information's mean value to solve the problem as a deterministic one [10]. The interval numbers are a natural, simple, and effective approach to express imperfect knowledge. This paper incorporates interval analytics to express the parameters of a preferential model.

On the other hand, when we apply MOEAs to solve problems with many objectives, they face challenges such [2–4]:

1. The exponential growth of the number of non-dominated solutions, making it harder to obtain representative samples of the Pareto front.
2. The increase in the number of poor solutions that are difficult to dominate (at least one of your objectives has a value, and the rest are close to optimal).
3. The solutions in the variable space become more distant as more objectives are added to the problem [11]. In such a case, when two distant parent solutions are recombined, the generated offspring solutions likely are also distant [12]; therefore, the efficiency of the genetics operators is questionable.
4. The high computational cost to determine the degree of diversity of the population.

Even though incorporating preferences in MOEAs is a challenging problem, the outranking approach handles it appropriately and aids in reducing the DM's cognitive effort required to choose a final solution [13]. Considering the lack of research devoted to studying the convenience of using the outranking approach in the optimization process, this work proposes a further analysis to observe the performance of a novel strategy of incorporating outranking in a MOEA. Unlike Cruz et al. [6], which requires representative solutions of two classes from the DM, this work proposes to incorporate two classes for internal use to guide the search process and establish greater differentiation between solutions, exerting selective pressure to find the ROI, but with the same cognitive load for the DM.

According to the reviewed literature [2–4,11], and as was mentioned before, MOEAs present difficulties when the number of objectives grows. For example, the classical Non-dominated Sorting Genetic Algorithm II (NSGA-II) [14] presents issues with the diversity-controlling operators [12]; authors extended this algorithm in NSGA-III to replace the crowding distance operator with the generation of well-spread reference point. In this paper, we propose a new method to integrate the DM's preferences to NSGA-III, which can deal with many objectives and is based on non-dominated fronts' ordering.

To the best of our knowledge, few of the previous studies has incorporated the presence of imperfect knowledge, nor have used the INTERCLASS-nC [15] as a classifier in the non-dominated-sorting process or employed more of two of inner classes to guide the search process towards the region of interest, and this work focuses on these issues. This research seeks to evaluate the proposed method's performance when incorporating preferences in the presence of imperfect knowledge with various versions of the proposed algorithm.

The remain of this paper is organized as follows—Section 2 includes reviewing the literature and some definitions of INTERCLASS-nC. Section 3 details the proposed method present. Section 4 specifies the benchmark to be solved, which includes seven problem instances. Section 5 shows and discusses the experimental results. Finally, Section 6 presents the conclusions of this paper and future work.

## 2. Literature Review

Two main approaches are distinguished in the area of Multi-Criteria Decision-Making (MCDM) [16]:

a  The French approach, based on outranking relationships built through comparisons between pairs of solutions to determine, for each pair of solutions, if there is relevant information (preference, indifference, or incompatibility) among them.

b  The American Multi-Attribute Utility Theory (MAUT) works based on the formulation of an overall utility function, and an interactive process can obtain this.

In the case of outranking relationship, indicators of dominance or preference are defined given some thresholds. This approach's main criticism is the difficulty to obtain the model parameters [6]; however, there are methods to solve it [17]. On the other hand, MAUT does not work when intransitivity exists between the preferential model [16]. The intransitivity phenomenon occurs in many real cases when exist a looping between the alternatives to select. It is important to consider this property to avoid possible incoherent solutions [18].

The incorporation of *interactive* and *a priori* preferences can reduce the search space because the information is used to guide MOEAs to reach the ROI, which is the region of the Pareto frontier preferred by the DM's. Expressing a DM's preference could be a more difficult cognitive process. According to Cruz et al. [6], the following characteristics are desirable for a preference incorporation method:

1.  Easy interaction between the DM and the solution algorithm involves minimizing the cognitive effort of a DM when making a judgment about the solutions.

2.  There should be no requirement for comparability and transitivity of preferences.

3.  The preference aggregation model must be compatible with the relevant characteristics of the real DMs.

4.  There should be techniques to infer the decision model parameters from examples provided by the DM.

In Cruz et al. [6], the multicriteria ordinal classification requires the DM to separate solutions into two categories. In a preference incorporation method with this classifier, the human categorization is the stage with the lowest cognitive demand of the entire process. Assigning solutions to the class "good" or "not good" does not require the DM to worry about the transitivity between them in the same way; the DM only compares the solutions between "good" and "not good".

Using outranking relationships allows handling the characteristics of many DMs facing real-world problems [6]. Being good that used preference incorporation methods meet the desirable characteristics described above, related to interaction with the DM, compatibility between the preferential model and the DM, properties of the preferences, and parameters' inference.

The ordinal multi-criteria classification can be useful to the DM to determine the best solution of a discrete set of alternatives, this is due to the existence of ordinally ordered sets starting with the most preferred alternatives to the least preferred ones [19]. There is a variety of multi-criteria ordinal classification methods, these can be grouped into the following classes [15]:

a  Methods based on the objective function value.

b  Symbolic methods, mainly those belonging to the theory of rough sets.

c  Methods based on outranking relationships.

To our knowledge, the first article that uses multi-criteria ordinal classification based on outranking was Oliveira et al. [20], which uses the popular ELECTRE-TRI method for ordinal classification in a three-objective problem, in which preferences are incorporated *a priori*, directly setting the parameters of the outranking model. Those methods belong to the family ELECTRE (*Elimination Et Choix Traduisant la Realite*) which uses a relation of outranking to identify if a solution *x* is at least as good as a *y*.

The hybrid algorithm proposed by Cruz et al. [13] uses a multi-criteria ordinal classification based on outranking. During the first phase, a meta-heuristic algorithm obtains a first approximation to the Pareto frontier. In the second phase, the DM assigns the solutions to two ordered classes and obtains the parameters of the outranking model. In the third phase, the THESEUS classification method applies selective pressure towards "satisfactory" solutions. They test the proposal on project portfolio problems with 4, 9, and 16 objectives; its results surpass the popular NSGA-II and Non-Outranked Ant Colony Optimization (NOACO) proposed in [21].

Cruz et al. [6] proposed the Hybrid Evolutionary Algorithm guided by Preferences (HEAP) algorithm, an extension of their previous work [13]. Where, instead of NSGA-II and NOACO, they use MOEA/D and MOEA/D-DE as metaheuristics for the first phase of the hybrid algorithm. For evaluating the proposed algorithm, they used instances of the portfolio optimization problem and the scalable test DTLZ problem, with three and eight objectives. The DTLZ benchmark are box-constrained continuous n-dimensional multi-objective problems, scalable in fitness dimension. This experimentation aims to analyze different in the activation of classification and the restart of solutions. The use of the DTLZ test suite makes possible assess the closeness to the ROI of a DM and compare the performance with three and eight objectives. The DM's preferences are simulated through an outranking model. In addition to the THESEUS classification method, the popular ELECTRE-TRI is incorporated, and the results of both methods are compared. In most cases, the best results were obtained with ELECTRE-TRI.

Additionally, few of the researches in the state of the art consider the imperfect knowledge in the DM's preferences and its effect in the function's objectives to be optimized. Besides, none has used the classifier INTERCLASS-nC in the non-dominated-sorting process or employed more inner classes to guide the search process towards the ROI. The proposed NSGA-III-P incorporates these characteristics.

### 2.1. Interval Arithmetic

In [22], Moore et al. formally proposed the interval analysis. An interval number can be viewed as an entity that reflects a quantitative property whose precise value is unknown. Still, the range within the value lies is known [15]. In this work, the imperfect knowledge is represented with interval numbers, Moore et al. [23] describes a number in interval as a range, $\mathbf{E} = [\underline{E}, \overline{E}]$, where $\underline{E}$ represents the lower limit while $\overline{E}$ the upper limit of an interval. Items in bold are numbers in intervals.

Considering two numbers of intervals $\mathbf{D} = [\underline{D}, \overline{D}]$ and $\mathbf{E} = [\underline{E}, \overline{E}]$, the Basic arithmetic operations can be defined for numbers of intervals as follows:

- addition:

$$\mathbf{D} + \mathbf{E} = [\underline{D} + \underline{E}, \overline{D} + \overline{E}] \tag{1}$$

- subtraction:

$$\mathbf{D} - \mathbf{E} = [\underline{D} - \overline{E}, \overline{D} - \underline{E}] \tag{2}$$

- multiplication:

$$\mathbf{D} * \mathbf{E} = [\ \min\{\underline{DE}, \underline{D}\overline{E}, \overline{D}\underline{E}, \overline{DE}\},\ \max\{\underline{DE}, \underline{D}\overline{E}, \overline{D}\underline{E}, \overline{DE}\}\ ] \tag{3}$$

- division:

$$\mathbf{D/E} = [\underline{D}, \overline{D}] * [\frac{1}{\underline{E}}, \frac{1}{\overline{E}}]. \tag{4}$$

According to Fliedner et al. [24] a *realization* of an interval number is any real number $e \in [\underline{E}, \overline{E}]$. An order relation is defined in the number of intervals as: let $e$ and $d$ be two realizations of $\mathbf{E}$ and $\mathbf{D}$ respectively, we say that $\mathbf{E} > \mathbf{D}$ if the preposition "*e is greater than d*" has greater credibility than "*d is greater the an e*".

Fernandez et al. [25] proposes the possibility function:

$$P(\mathbf{E} \leq \mathbf{D}) = \begin{cases} 1 \text{ if } p_{ED} > 1, \\ P_{ED} \text{ if } 0 \leq P_{ED} \leq 1, \\ 0 \text{ if } P_{ED} < 0, \end{cases} \tag{5}$$

where $\mathbf{E} = [\underline{e}, \overline{e}]$ and $\mathbf{D} = [\underline{d}, \overline{d}]$ are numbers of intervals and $P_{ED} = \frac{\overline{e} - \underline{d}}{(\overline{e} - \underline{e}) + (\overline{d} - \underline{d})}$. The order relationship between $\mathbf{D}$ and $\mathbf{E}$ is given by:

a     If $\underline{D} = \underline{E}$ and $\overline{D} = \overline{E}$, then $\mathbf{D} = \mathbf{E}$. Therefore $P(\mathbf{E} \geq \mathbf{D}) = 0.5$.

b     If $\underline{E} > \overline{D}$, then $\mathbf{E} > \mathbf{D}$. Therefore $P(\mathbf{E} \geq \mathbf{D}) = 1$.

c     If $\overline{E} < \underline{D}$, then $\mathbf{E} < \mathbf{D}$. Therefore $P(\mathbf{E} \geq \mathbf{D}) = 0$.

d     If $\underline{D} \leq \underline{E} \leq \overline{D} \leq \overline{E}$ or $\underline{D} \leq \underline{E} \leq \overline{E} \leq \overline{D}$, when:

      (a)     $P(\mathbf{E} \geq \mathbf{D}) > 0.5$. Therefore, $\mathbf{E}$ is greater than $\mathbf{D}$, $(\mathbf{E} > \mathbf{D})$.

      (b)     $P(\mathbf{E} \geq \mathbf{D}) < 0.5$. Therefore, $\mathbf{E}$ is less than $\mathbf{D}$, $(\mathbf{E} < \mathbf{D})$.

### 2.2. INTERCLASS-nC

Fernandez et al. [15] proposed an ordinal classification method, useful when the DM has a vague idea about the boundaries between adjacent classes but can identify several (even one) representative solutions in each class.

The DM must provide a model of outranking in terms of:

- Weight, $\mathbf{w} = [w^-, w^+]$
- Veto threshold, $\mathbf{v} = [v^-, v^+]$
- Majority threshold $\lambda = [\lambda^-, \lambda^+]$
- Credibility threshold $\beta = [\beta^-, \beta^+]$.

A set of classes $C = \{C_1, ..., C_k, ..., C_m\}$, $(m \geq 2)$ is defined, ordered by increasing preference. Considering a $\delta > 0.5$ and $\lambda > [0.5, 0.5]$. Where, $\delta$ corresponds to the maximum probability degree for which the strength of the coalition of agreement exceeds $\lambda$.

$R_k = \{r_{kj}, j = 1, ..., card(R_k)\}$ is a subset of reference solutions that characterize $C_k, k = 1, ..., m$ and $\{r_0, R_1, ..., R_m, r_{m+1}\}$ is the set of all reference solutions, in which $r_0$ and $r_{m+1}$ are the worst and the ideal reference solution respectively. The elements in $R_k, k = 1, ..., m - 1$ must satisfy the conditions defined in Fernandez et al. [15].

Classification is performed using top-down and bottom-up methods jointly. Each method proposes a class for the assignment of $x$; in case of not coinciding, these rules propose a possible range for the assignment of $x$.

### 3. Proposed Method

The Nondominated Sorting Genetic Algorithm III proposed in [12] is a genetic algorithm similar to the original NSGA-II. They search the Pareto optimal set performing a non-dominated sorting. The difference is the maintenance of diversity in the selection stage. The first uses crowding distances, and the second uses reference points. NSGA-III discriminates between the non-dominated solutions using a utility function, which calculates a solution's relevance to approximate a reference point.

To incorporate a DM's preferences, we propose integrating the ordinal classification method INTERCLASS-nC into the NSGA-III, we will call this variant NSGA-III-P. The original work [6] only defines the classes "satisfactory" (Sat) and "unsatisfactory" (Dis); the DM gives a reference set to generate these classes (with one or more representative solutions for each class). This classification complements the non-dominated sorting to increase the capacity to discriminate solutions; this strategy induces a greater selective

pressure, focusing the search toward the ROI. In this work, two classes are added internally for giving more precision in the comparison of the solutions:

- The DM is highly satisfied (*HSat*) with an $x$ solution, if for each action $w \in R_2$ it is true that $xPr(\beta, \lambda)w$.
- The DM is highly dissatisfied (*HDis*) with an $x$, if for each action $w \in R_1$ it is true that $wPr(\beta, \lambda)x$.

The steps to follow to generate the $P_{t+1}$ of the NSGA-III-P that integrates the INTERCLASS-nC ordinal classification method are shown in the Algorithm 1. Let $Q_t$ the children population of the current generation with equal number of individual $N$ of $P_t$. The first step is to combine the children and parents tending $R_t = P_t \cup Q_t$ (of size $2N$), the $N$ individuals that will become $P_{t+1}$ will be selected. To do this, $R_t$ will be divided into multiple fronts not dominated by *non-dominated sorting* $(F_1, F_2, ..., F_n)$.

The proposed method of integration of preferences works with the set of previously created non-dominated fronts, by classifying all the solutions in $F_1$ and group the solutions in classes, creating the fronts $F'_1, F'_2, F'_3, F'_4$ corresponding to classes *HSat, Sat, Dis, HDis*. In the created fronts are joined with the remaining ones in such a way that $F' = \{F'_1, F'_2, F'_3, F'_4\} \cup_{j=2}^n F_j$. This process is illustrated in Figure 1 and corresponds to step 7–18 in Algorithm 1.

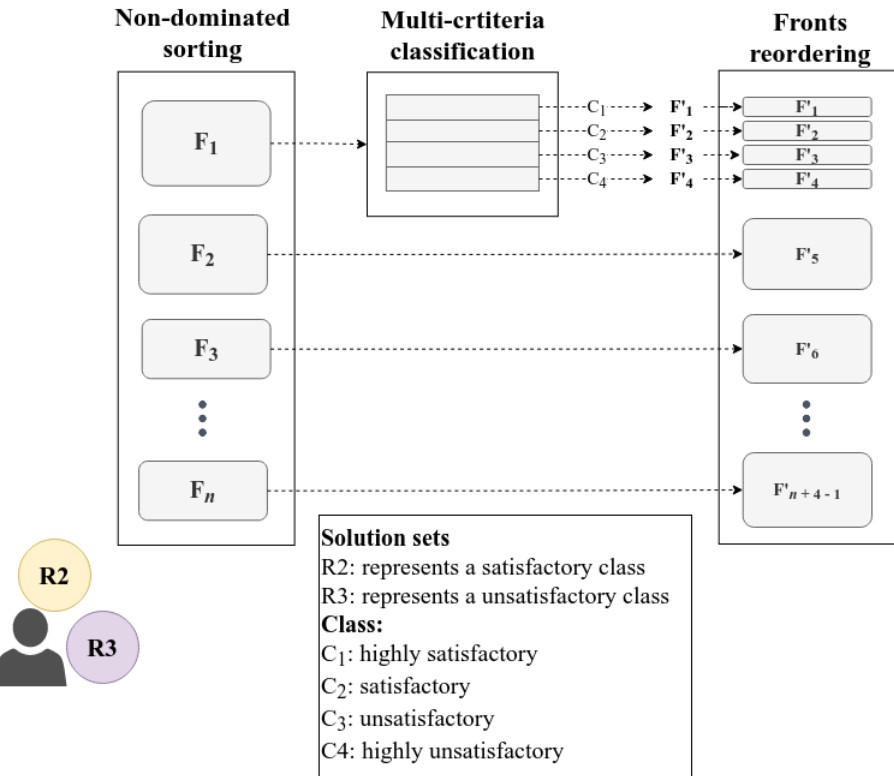

**Figure 1.** The proposed methodology for classifying the $F_1$, grouping, and fronts reordering.

After $F'_1$ the new population is built until the size is $N$. The last front is called the $l$-th front. Therefore, the front $l + 1$ are rejected; in most situations, $l$ is partially accepted. Only the solutions that maximize the diversity of $l$-th are selected in such a case (steps 21–26).

---

**Algorithm 1** Generation $P_t$ of NSGA-III-P

---

**Input:** $H$ structured reference points $Z^s$ or supplied aspiration points $Z^a$, parent population $P_t$, $Cx$ iteration where the algorithm applies the classification, $Ry$ solution replacement rate

**Output:** $P_{t+1}$

1:   $S_t \leftarrow \varnothing, i \leftarrow 1$
2:   $Q_t \leftarrow Recombination + Mutation(P_t)$
3:   $R_t \leftarrow P_t \cup Q_t$
4:   $(F_1, F_2, ..., F_n) \leftarrow Non-dominated-sort(R_t)$
5:   $//$ If the rest of the current *iteration* between $Cx$ equals 0, the classification applies
6:   **if** $(iteration \ mod \ Cx) == 0$ **then**
7:      $(F_1', F_2', F_3', F_4') \leftarrow \varnothing$
8:      **for** $s \in F_1$ **do** $//$ Classify each member of $F_1$ and group by class
9:          $c \leftarrow classify(s)$
10:          **if** $c == "hsat"$ **then**
11:              $F_1' \leftarrow F_1' \cup s$
12:          **if** $c == "sat"$ **then**
13:              $F_2' \leftarrow F_2' \cup s$
14:          **if** $c == "dis"$ **then**
15:              $F_3' \leftarrow F_3' \cup s$
16:          **if** $c == "hdis"$ **then**
17:              $F_4' \leftarrow F_4' \cup s$
18:      $F' \leftarrow \{F_1', F_2', F_3', F_4'\} \cup_{j=2}^{n} F_j$ $//$ Fronts reordering
19:   **else**
20:      $F' = (F_1, F_2, ..., F_n)$
21:   **while** $|S_t| \leq N$ **do** $//$ Last front to be included $F_l' \leftarrow F_i'$
22:      $S_t \leftarrow S_t \cup F_i'$
23:      $i \leftarrow i + 1$
24:   **if** $|S_t| == N$ **then**
25:      **if** $(iteration \ mod \ Cx) == 0$ **then**
26:          $replacement(S_t, Ry)$ $//$ Replace the last $Ry$ random individuals
27:      **Return:** $S_t$
28:   **else**
29:      $P_{t+1} \leftarrow \cup_{j=1}^{l-1} F_j$
30:      Points to be chosen from $F_l : K \leftarrow N - |P_{t+1}|$
31:      Normalize objectives & create reference set $Z^r \leftarrow normalize(f^n, S_t, Z^r, Z^s, Z^a)$
32:      Associate each member $s \in S_t$ with a reference point:
33:      $[\pi(s), d(s)] = associate(S_t, Z^r) \% \pi(s)$
34:      Compute niche count of reference point $j \in Z^r : p_j = \sum_{s \in S_t/F_l}((\pi(s) = j)?1 : 0)$
35:      Choose $K$ member one at a time from $F_l$ to construct
36:      $P_{t+1} : niching(K, p_j, \pi, d, Z^r, F_k, P_{t+1})$
37:      **if** $(iteration \ mod \ Cx) == 0$ **then**
38:          $replacement(S_t, Ry)$ $//$ Replace the last $Ry$ random individuals
39:      **Return:** $P_t$

---

The proposed algorithm has two approaches for controlling the selective pressure generated by the incorporation of preference:

- Apply classification every certain number of iterations (step 6).
- Incorporating a replacement mechanism of $Ry$ individuals from the population (steps 25 and 36), this criterion only applies when classification occurs.

Preference incorporation is, in a certain way, an Intensification approach. The Intensification would be reduced by adding new random solutions and generating a diversification, therefore balancing the search. We analyzed different activation configurations in the experimental section to observe their impact on the algorithm's performance.

## 4. Experimental Settings

The proposed NSGA-III-P (non-nominated sorting genetic algorithm III with preferences) algorithm's experimentation was carried out to solve the DTLZ1 - DTLZ7 problem's. The algorithm's performance is observed to evaluate the effect of the intensification-diversification mechanism.

All the algorithms used in this experimentation were executed 50 times for each instance on an Intel Core i7-10510U CPU @ 1.80GHz × 8 with 16 GB of RAM. We developed the algorithms in Java using the OpenJDK 11.0.10 64-Bit.

The DTLZ problem's instances configuration is summarized in the Table 1. For his solution, the algorithm has a population size $n = 92$ individuals, the algorithm uses the SBX crossover operator and the polynomial mutation operator. The Table 2 shows the configurations of these operators.

**Table 1.** Parameters Used for Three-Objective DTLZ Problem's instances.

| Problem | Number of Variables | Iterations |
|---|---|---|
| DTLZ1 | 7 | 400 |
| DTLZ2 | 12 | 250 |
| DTLZ3 | 12 | 1000 |
| DTLZ4 | 12 | 600 |
| DTLZ5 | 12 | 500 |
| DTLZ6 | 12 | 500 |
| DTLZ7 | 12 | 500 |

**Table 2.** Crossover and mutation parameters used for NSGA-III-P.

| Parameter | Value |
|---|---|
| Polynomial mutation probability $p_m$ | $\frac{1}{n}$ |
| Polynomial mutation index $n_m$ | 20 |
| SBX crossover probability $p_c$ | 1 |
| SBX crossover index $n_c$ | 30 |

We analyzed the NSGA-III-P algorithm's versions named $CxRy$, where $x$ is the percentage of iterations to activate the classification. In contrast, $y$ is the percentage of replacement of solutions. Considering the classification increase intensification, less classification reduces the intensification, and restart of solutions increases the diversification; these variants are higher to lower intensification: C100R0, C1R0, C1R2, C10R0, and C0R0 (see Table 3).

**Table 3.** Experimental configurations carried out.

| Name | Description |
|---|---|
| **C0R0** | NSGA-III reported in the literature. |
| **C100R0** | NSGA-III-P with classification in each iteration with 0% replacement. |
| **C10R0** | NSGA-III-P with classification every 10% iterations with 0% replacement. |
| **C1R0** | NSGA-III-P with classification every 1% iterations with 0% replacement. |
| **C1R2** | NSGA-III-P with classification every 1% iterations with 2% replacement. |

### 4.1. Creation of the ROI

Let $T'$ be a sample of non-dominated solutions taken from a large set $T$ of solutions ($\geq 100$ thousand) generated analytically at the Pareto frontier of a standard problem. The solutions that integrate the ROI identified with the following sets and measures in $T'$.

- Outranking weakness of a solution $x$. A low value of this measure provides positive arguments for selecting $x$.

$$D_o(x) = \{y | \sigma(y, x) > \beta, \ \sigma(x, y) < 0.5, \ y \in T'\{x\}\} \tag{6}$$

- Net score measure used to identify DM preferred solutions.

$$F_n(x) = \sum_{y \in T'} \sigma(x, y) - \sigma(y, x) \tag{7}$$

where $F_n(x) > F_n(y)$ indicates a certain preference of $x$ over $y$.
- Best compromise solution set more preferred by the DM.

$$x^* = \{x | D(x) = 0, F_n(x) = max_{y \in T'}(F_n(y)), x \in T'\} \tag{8}$$

- Region of interest made up of the best compromise solutions $x^*$

$$ROI(T') = x^* \cup \{max_{x \in T'}(F_n(x) \geq 0, K)\}, \tag{9}$$

where $K$ are the largest $F_n$ values of $x$.

### 4.2. Indicators of Performance

Each algorithm is executed 50 times to the result of a complete run of the NSGA-III-P algorithm configurations, and applying the following indicators:

a    Minimum, mean, and maximum Euclidean distance among the obtained non domi­nated solutions and the ROI (also called Min Euclid, Mean Euclid, Max Euclid)
b    Conservation of Dominance: creates a set of non-dominated solutions from the solu­tions obtained from all configurations. Counting the solutions of each configuration.
c    Conservation of Satisfaction: the non-nominated solutions belonging to the HSat and Sat classes (classified by the INTERCLASS-nC) are counted.

### 4.3. Description of the Instance

The DTLZ problems instance used contains the characterization of the DM preferences (elements 3–6). It has the following elements:

1.    objectives number: integer
2.    variable number: integer
3.    weight vector: *Interval*
4.    veto vector: *Interval*
5.    lambda: *Interval*
6.    references solutions: a vector of solutions is expected.

## 5. Results

Table 4 shows the reached performance for each algorithm when solving each DTLZ problem. For space reasons, these results are only presented for two performance mea­sures. The first two columns show the result for the original NSGA-III algorithm. The next columns present eight variants of NSGA-III with preferences. The first six columns corre­spond to variants without activating the solutions restarting strategy. The last two columns correspond to variants that use restarting to reduce the effect of incorporate preferences.

**Table 4.** Average algorithm performance evaluated with two measures for DTZL problems.

| Problem | NSGA-III | | NSGA-III-P (with Preferences) | | | | | | | |
| | | | without Restart | | | | | | with Restart | |
| | C0R0 | | C100R0 | | C10R0 | | C1R0 | | C1R2 | |
| | %C CHSat | Min Euc | %C CHSat | Min Euc | %C CHSat | Min Euc | %C CHSat | Min Euc | %C CHSat | Min Euc |
|---|---|---|---|---|---|---|---|---|---|---|
| DTLZ 1 | $1.946^{3.5}$ | $0.007456^{3.0}$ | $\mathbf{92.769}^{1.0}$ | $0.001912^{3.0}$ | $1.589^{3.5}$ | $0.005215^{3.0}$ | $1.565^{3.5}$ | $0.010437^{3.0}$ | $2.131^{3.5}$ | $0.011874^{3.0}$ |
| DTLZ 2 | $0.843^{3.5}$ | $0.007459^{3.0}$ | $\mathbf{97.260}^{1.0}$ | $0.003802^{3.0}$ | $0.625^{3.5}$ | $0.008952^{3.0}$ | $0.604^{3.5}$ | $0.014182^{3.0}$ | $0.668^{3.5}$ | $0.013754^{3.0}$ |
| DTLZ 3 | $8.434^{3.5}$ | $0.029905^{5.0}$ | $\mathbf{76.477}^{1.0}$ | $0.029269^{3.0}$ | $5.524^{3.5}$ | $0.067064^{3.0}$ | $4.481^{3.5}$ | $0.049607^{5.0}$ | $5.083^{3.5}$ | $0.064364^{5.0}$ |
| DTLZ 4 | $2.661^{3.5}$ | $0.000131^{3.0}$ | $\mathbf{78.974}^{1.0}$ | $0.000001^{3.0}$ | $2.567^{3.5}$ | $0.000001^{3.0}$ | $11.092^{3.5}$ | $0.000796^{3.0}$ | $4.706^{3.5}$ | $0.000002^{3.0}$ |
| DTLZ 5 | $0.365^{3.5}$ | $0.001888^{3.0}$ | $\mathbf{56.065}^{1.0}$ | $0.001635^{3.0}$ | $0.852^{3.5}$ | $0.005414^{3.0}$ | $25.549^{3.5}$ | $0.000864^{3.0}$ | $17.169^{3.5}$ | $0.000819^{3.0}$ |
| DTLZ 6 | $1.259^{3.5}$ | $0.004893^{3.0}$ | $\mathbf{52.887}^{1.0}$ | $0.0009437^{5.0}$ | $1.614^{3.5}$ | $0.004585^{5.0}$ | $23.793^{3.5}$ | $0.001554^{5.0}$ | $20.446^{3.5}$ | $0.001213^{3.0}$ |
| DTLZ 7 | $12.148^{3.5}$ | $0.039196^{3.5}$ | $\mathbf{42.988}^{1.0}$ | $\mathbf{0.006155}^{1.0}$ | $11.044^{3.5}$ | $0.039166^{3.5}$ | $17.744^{3.5}$ | $0.027163^{3.5}$ | $16.075^{3.5}$ | $0.028475^{3.5}$ |
| Average | 3.95086 | 0.01299 | 71.06 | 0.00624 | 3.40214 | 0.01863 | 12.11829 | 0.03318 | 9.46829 | 0.01827 |

%C-CHSat: conservation percentage of highly satisfactory solutions; MinEuc: min Euclidean distance.

Table 5 shows the first summary of a statistical comparison of five variants of NSGA-III using the configurations reported in Table 4. We applied the Friedman Test, followed by the Hollman Post-hoc Test. The best and the worst algorithm are identified with the algorithms' ranking considering two measures: the percentage of conservation of highly satisfactory solutions (CHSat) and the minimum Euclidean distance (MinEuc).

**Table 5.** Best and worst algorithms resulting from their statistical comparison evaluated with two measures.

| PROBLEM | Best Variants for | | Worst Variants for | |
| | CHSat | Min | CHSat | Min |
|---|---|---|---|---|
| DTLZ1 | C100R0 | C0R0 | C1R0 | C1R2 |
| DTLZ2 | C100R0 | C100R0 | C1R0 | C1R2 |
| DTLZ3 | C100R0 | C100R0 | C1R0 | C10R0 |
| DTLZ4 | C100R0 | C100R0 | C10R0 | C1R2 |
| DTLZ5 | C100R0 | C1R2 | C0R0 | C10R0 |
| DTLZ6 | C100R0 | C1R2 | C0R0 | C0R0 |
| DTLZ7 | C100R0 | C100R0 | C10R0 | C0R0 |

In this paper, the main measure to evaluate algorithms is related to the counting of highly satisfactory solutions because preferences elicitation is aligned with this measure. But considering other DM could be interested in the closeness to the ROI, the Euclidean distance is an alternative because it is frequently used in decision-making. For a DM interested in highly satisfactory solutions, the best variant for all DTLZ problems is C100R0. In contrast, if the DM is interested in solutions closer to the ROI, we cannot find a unique variant as the best; They are dependent on the problem. The C100R0 variant offers solutions close to the ROI in four of the seven problems evaluated (DTLZ2–DTLZ4, DTLZ7); For the DTLZ5 and DTLZ6 problems, C1R2 has a better performance. The original NSGA-III algorithm offers solutions closer to the ROI for the DTLZ1 problem. It is noteworthy. that C100R0 is never the worst option; the other variants are the worst at least once.

Table 6 shows the algorithms' average performance for all DTLZ problems. After applying statistical tests to compare algorithms (Friedman aligned and Hollman posthoc). We identify pairwise comparisons with significant differences. Using these pairs, for each algorithm, a set of statistically no better algorithms was obtained. Finally, the algorithms are ranked instead of Hierarchical using the well-known Borda count to accumulate their positioning overall instances for a given measure. The superscript corresponds to ranking Borda.

There are significant statistical differences in 3 of the 5 metrics evaluated (CHSAT, Mean Euclidean, Max Euclidean). For the percentage of conservation of solutions for which

the DM is highly satisfied (CHSat), the best algorithm is C100R0. In contrast, the rest of the algorithms have a similar behavior according to Borda's ranking. The indicator of the percentage of solutions for which the DM is satisfied (CSat) does not significantly differ. That is expected because CHSAT gets better well-solutions.

**Table 6.** The average and standard deviation of the algorithms over 50 independent runs in terms of percentage of conservation and Euclidean distance for the DTLZ family of problems.

| Configuration | % of Conservation | | Euclidean Distance | | |
| | CHSat | CSat | Min | Mean | Max |
| --- | --- | --- | --- | --- | --- |
| C0R0 | $4.659^{2.0}_{0.124}$ | $4.877^{3.0}_{0.053}$ | $0.002289^{2.0}_{0.003}$ | $0.779536^{4.0}_{0.396}$ | $2.643408^{1.0}_{2.801}$ $\uparrow$ |
| C100R0 | $62.169^{1.0}_{0.259}$ $\uparrow$ | $6.736^{3.0}_{0.105}$ | $0.000723^{1.0}_{0.001}$ $\uparrow$ | $0.179397^{1.0}_{0.095}$ $\uparrow$ | $0.724694^{1.0}_{0.448}$ $\uparrow$ |
| C10R0 | $4.239^{2.0}_{0.109}$ | $7.355^{2.0}_{0.061}$ | $0.001157^{2.0}_{0.001}$ | $0.790513^{4.0}_{0.403}$ | $1.526600^{2.0}_{0.881}$ |
| C1R0 | $15.951^{2.0}_{0.279}$ | $38.098^{1.0}_{0.356}$ $\uparrow$ | $0.001337^{3.0}_{0.002}$ | $0.676294^{2.0}_{0.410}$ | $1.447190^{2.0}_{0.935}$ |
| C1R2 | $12.981^{2.0}_{0.230}$ | $42.934^{1.0}_{0.424}$ $\uparrow$ | $0.002135^{2.0}_{0.005}$ | $0.697033^{3.0}_{0.422}$ | $1.497078^{2.0}_{0.899}$ |

Statistical test: Friedman of aligned ranks with a significance level of 0.05. The superscript indicates the position in which it was ranked by the Borda method. The subscript indicates the standard deviation of the results. The upper arrow indicates the top-ranked algorithm.

The C100R0 configuration is the one with the greatest contribution of solutions closer to the ROI according to the minimum Euclidean distance indicator. This indicator does not have significant differences. For the average, significant differences were found, and the algorithm C100R0 is the one that provides the closest solutions. The algorithms that provide the least distant solutions are C100R0 and C0R0 based on the maximum of the Euclidean distance.

This global analysis gives the best rank for the C100R0, meaning that it is a good alternative for all analyzed problems. However, C1R2 produces solutions closer to the ROI in some problems. They are extreme variants concerning intensification and diversification, meaning that the balance between them depends on the problem; we need to conduct extensive experimentation to confirm.

To illustrate the superiority of the proposed NSGA-III-P concerning NSGA-III, Figures 2 and 3 shows the non-dominated solutions obtaining when solving the DTLZ3 problem. Figure 2 is for NSGA-III (C0R0) and Figure 3 is for NSGA-III-P with preferences all time and without a restart (C100R0). The variant C100R0 performs a better exploration of the region of interest with highly satisfactory solutions. At the same time, C0R0 scans the entire solution space, but most solutions are highly unsatisfactory. The solutions belonging to the ROI are illustrated in black, the solutions classified as highly satisfactory (HSat) in green, satisfactory solutions (Sat) in blue, unsatisfactory solutions orange (Dis), and highly unsatisfactory solutions (HDis) in red.

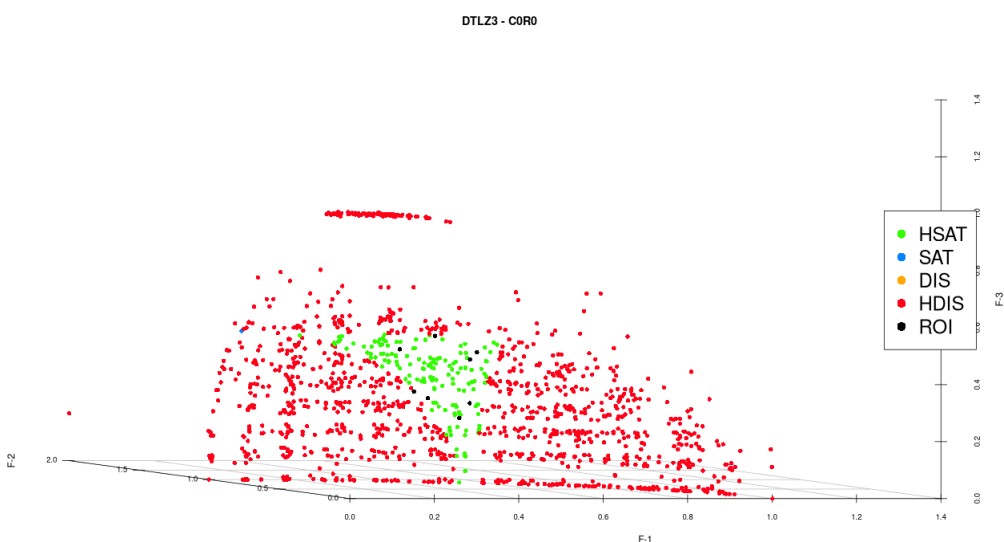

**Figure 2.** Non-dominated NSGA-III(C0R0) solutions of the DTLZ3 problem.

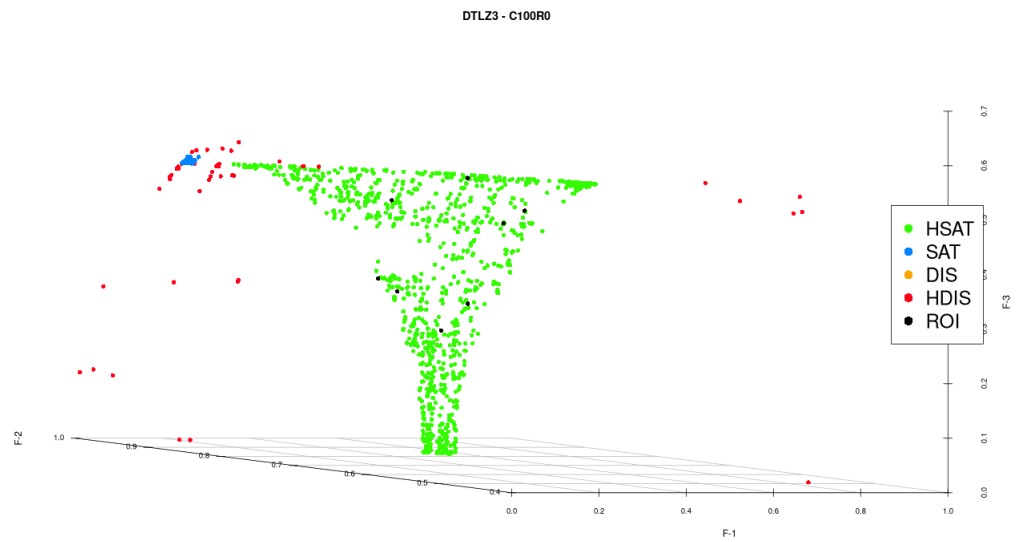

**Figure 3.** Non-dominated NSGA-III-P(C100R0) solutions of the DTLZ3 problem.

## 6. Conclusions

This article presents a novel method for incorporating DM's preferences into the NSGA-III algorithm, named NSGA-III-P. INTERCLASS-NC is a multi-criteria and outranking ordinal classifier that allows incorporating preference, giving the algorithm the capacity to improve the discrimination of solutions and intensify the search toward the region of interest. Excessive intensification can diminish the algorithm's effectiveness. To regulate this selective pressure, we add two complementary strategies to the search in NSGA-III-P: control the activations of the classification and control the restarts of solutions.

Experiments with different configurations of NSGA-III-P were proposed to study different levels of intensification and diversification. NSGA-III-P solve the DTLZ test suite, including the preferences of DM with imperfect knowledge.

Based on computational experimentation, the best alternative to the DTLZ problems is the C100R0 (always classify without restarts) when the DM is looking for highly satisfactory solutions. When the DM prefers solutions closer to the ROI, the variants C1R2 (classify and sometimes restart) and C100R0 have the best performance with two and four problems,

respectively. In general, the proposed method NSGA-III-P outperforms NSGA-III because it allows obtaining better approximations to the ROI using the principal performance measures; only in one case, the NSGA-III is the best option for the DTLZ1 problem using the Max Euclidean distance.

These preliminary results open a research line to determine the extent to which the selective pressure induced by preferences improves the algorithm performance concerning the closeness to the ROI and the factors that affect it.

As future work, we will evaluate the proposal with a greater number of objectives for the DTLZ problems. Also, the proposal will be integrated into at least one other algorithm representative of the state of the art. We aim to develop a method that dynamically adjusts the diversification and intensification levels required for each problem.

**Author Contributions:** Conceptualization, A.C.-A., L.C.-R. and E.F.; methodology, L.C.-R., N.R.-V. and A.C.-A.; software, A.C.-A. and N.R.-V.; validation, L.C.-R., N.R.-V., H.F., J.A.B.-H., C.G.-S. and A.C.-A.; formal analysis, L.C.-R.; investigation, L.C.-R., E.F. and A.C.-A.; resources, E.F., L.C.-R. and A.C.-A.; data curation, N.R.-V. and A.C.-A.; writing—original draft preparation, A.C.-A. and L.C.-R., J.A.B.-H.; writing—review and editing, C.G.-S., L.C.-R., H.F., J.A.B.-H., N.R.-V., J.A.B.-H. and A.C.-A.; visualization, L.C.-R.; supervision, L.C.-R.; project administration, L.C.-R.; funding acquisition, L.C.-R., A.C.-A. and H.F. All authors have read and agreed to the published version of the manuscript.

**Funding:** This research received no external funding.

**Data Availability Statement:** The instances and other files used here are available at https://www.dropbox.com/sh/5wb8api8zdyjs8y/AAD11EQbI4P0lQgijvgfFC2qa?dl=0 (accessed on 29 March 2021).

**Acknowledgments:** Authors thanks to CONACYT for supporting the projects from (a) Cátedras CONACYT Program with Number 3058. (b) CONACYT Project with Number A1-S-11012 from Convocatoria de Investigación Científica Básica 2017–2018 and CONACYT Project with Number 312397 from Programa de Apoyo para Actividades Científicas, Tecnológicas y de Innovación (PAACTI), a efecto de participar en la Convocatoria 2020-1 Apoyo para Proyectos de Investigación Científica, Desarrollo Tecnológico e Innovación en Salud ante la Contingencia por COVID-19. (c) Alejandro Castellanos-Alvarez would like to thank CONACYT for the support number 1006467.

**Conflicts of Interest:** The authors declare no conflict of interest.

## Abbreviations

The following abbreviations are used in this manuscript:

| | |
|---|---|
| DM | Decision-Maker |
| ROI | Region of Interest |
| MOP | Multi-objective Optimization Problem |
| MCDM | Multi-criteria Decision-Making |
| MAUT | Multi Attribute Utility Theory |
| HSat | Highly Satisfactory |
| Sat | Satisfactory |
| Dis | Unsatisfactory |
| HDis | Highly Unsatisfactory |

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
