# Peer review of "A Method for Integration of Preferences to a Multi-Objective Evolutionary Algorithm Using Ordinal Multi-Criteria Classification"

_mca, doi:10.3390/mca26020027_

Round 1

Reviewer 1 Report

The paper includes an interesting approach to tackle the imprecision which is called imperfect knowledge, and the results and conclusions seems promising. However, the paper requires an extensive review of the english. I include a pdf file with some specific notes to be taken into account to improve the english and style.

It is recommended to change the passive voice to active voice. However, it is not mandatory to change every passive voice note.
Specific changes may be changed directly in the document. and comment as “changed” in the response file for all of those notes.
Other notes require a larger explanation and should be addressed in the response file.

Author Response

We are grateful for the reviewer's helpful comments and thank him for his careful reading of our work. We have considered these comments in the second version of our work. And we hope that we have met your concerns and expectations. Our responses to each of the reviewer's specific comments are listed below.

1.1 COMMENT: The paper includes an interesting approach to tackle the imprecision which is called imperfect knowledge, and the results and conclusions seems promising. However, the paper requires an extensive review of the english.

R.           FIXED, the manuscript was subject to careful revision to fixed grammar mistakes and to widely improve the writing style.

1.2 COMMENT: It is recommended to change the passive voice to active voice. However, it is not mandatory to change every passive voice not

R.              Thanks for the comment; after carefully revising our writing, the use of passive voice has been changed conveniently.

1.3 COMMENT: I include a pdf file with some specific notes to be taken into account to improve the english and style. Specific changes may be changed directly in the document. and comment as “changed” in the response file for all of those notes.

R.              Thanks to the reviewer for the observations. Every specific note has been CHANGED, and now all of them are included in the revised version of the manuscript.

1.4 COMMENT: Other notes require a larger explanation and should be addressed in the response file.

R.              DONE, the remaining notes indicated in the reviewed pdf that required a better explanation were taken into account, and were improved accordingly so that the fluency and content of the revised manuscript were improved. The specific details on how each of those notes were addressed are provided in the following comments, indicating the line were they were mention in the old version:

1.4.1 COMMENT:

(L2) It is unclear the nature of the imperfect knowledge concept in this sentence. I recommend explaining it more clearly or with other words.

(L4) Are you referring to the imperfect knowledge problem?

(L6) Passive voice

(L7) Problems or instances?

(L8) If you're proposing an NSGA III algorithm, you should state it in the previous sentence

R. We rewrite the abstract to attend all comments. The abstract rewritten is the following:

“Most real-world problems require the optimization of multiple objective functions simultaneously, which can conflict with each other. The environment of these problems usually involves imprecise information derived from inaccurate measurements or the variability in DMs judgments and beliefs, which can lead to unsatisfactory solutions. The imperfect knowledge can be present either in objective functions, restrictions, or decision maker's preferences. These optimization problems have been solved using various techniques such as multi-objective evolutionary algorithms (MOEAs). This paper proposes a new MOEA called NSGA-III-P (non-sorting genetic algorithm III with preferences). The main characteristic of NSGA-III-P is an ordinal multi-criteria classification method for preference integration to guide the algorithm to the region of interest given by the decision maker’s preferences. Besides, the use of interval analysis allows express preferences with imprecision. The experiments contrasted several versions of the proposed method with the original NSGA-III to analyze different selective pressure induced by the DM´s. In these experiments, the algorithms solved three-objectives instances of the DTLZ problem. The obtained results showed a better approximation to the region of interest for a DM when its preferences are considered.”

1.4.2 COMMENT: (L13) It is the same intro as the Abstract, should be modified.

R. We rewrite the corresponding paragraph. The new paragraph is the following:

“Many industrial domains are concerned with multi-objective optimization problems (MOPs), which in general have conflicting objectives to handle [1]. To solve optimally a MOPs is to find a set of solutions defined as Pareto optimal solutions. They represent the best compromise between the conflicting objectives. A promising alternative is solving MOPs with metaheuristics, like multi-objective evolutionary algorithms (MOEAs); they obtain an approximation of the Pareto optimal set. This approach solves the problem partially since now the decision-maker (DM) has to choose the best compromise solution, which satisfies his preferences, from the set of solutions obtained (non-dominated by each other). For practical reasons, the DM needs to choose one solution to implement it.”

1.4.3 COMMENT: (L29) Rewrite and explain carefully, because, to my understanding, the imperfect knowledge is a new contribution and should be addressed that way.

R. We rewrite the corresponding paragraph and add new references. The final result appearing in the manuscript is:

“In many real-world situations, the MOPs environment implicates imprecise information derived from inaccurate measurements or the variability in DM's judgments and beliefs. Not considering these imprecisions can lead to unsatisfactory solutions and, in consequence, to a poor choice between the existing alternatives due to imperfect knowledge of the problem [6]. Imprecise information may be present in different components of a MOP; for example, it can be either in objective functions, restrictions, or a decision maker's preferences. Obtaining the preferential model parameters is a difficult task that increases with the objective number, which is only possible when the handle of imprecision is allowed [Balderas, 2019]. The simplest approach to handling imprecise information is to estimate this information's mean value to solve the problem as a deterministic one [Talbi, 2009]. The interval numbers are a natural, simple and effective approach to express imperfect knowledge. This paper incorporates interval analytics to express the parameters of a preferential model.”

Talbi, E. G. (2009). Metaheuristics: from design to implementation (Vol. 74). John Wiley & Sons.

Balderas, F., Fernandez, E., Gomez-Santillan, C., Rangel-Valdez, N., & Cruz, L. (2019). An interval-based approach for evolutionary multi-objective optimization of project portfolios. International Journal of Information Technology & Decision Making, 18(04), 1317-1358.

1.4.4 COMMENT: (L38) Please include a brief explanation. (The ineffectiveness of genetic operators).

R. We rewrite the corresponding paragraph as follows:

“The solutions in the variable space become more distant as more objectives are added to the problem[9]. In such a case, when two distant parent solutions are recombined, the generated offspring solutions likely are also distant [10] therefore, the efficiency of the genetics operators is questionable.”

1.4.5 COMMENT: (L48-49) What kind of problems? Rewrite.

R. We describe the presented difficulties when the number of objectives grows, rewrite the corresponding paragraph, and add more references. The result is the following:

“According to the reviewed literature [2-4; Jaimes, 2015], and as was mentioned before, MOEAs presents difficulties when the number of objectives grows. For example, the classical algorithms NSGA-II [Deb, 2002] presents issues with the diversity-controlling operators [Deb, 2013], authors extended this algorithm in NSGA-III to replace the crowding distance operator with the generation of well-spread reference point. In this paper, we propose a new method to integrate the DM's preferences to NSGA-III, which can deal with many objectives and is based on non-dominated fronts' ordering.”

Jaimes, A. L., & Coello, C. A. C. (2015). Many-objective problems: challenges and methods. In Springer handbook of computational intelligence (pp. 1033-1046). Springer, Berlin, Heidelberg.

Deb, K., Pratap, A., Agarwal, S., & Meyarivan, T. A. M. T. (2002). A fast and elitist multiobjective genetic algorithm: NSGA-II. IEEE transactions on evolutionary computation, 6(2), 182-197.

Deb, K., & Jain, H. (2013). An evolutionary many-objective optimization algorithm using reference-point-based nondominated sorting approach, part I: solving problems with box constraints. IEEE transactions on evolutionary computation, 18(4), 577-601.

1.4.6 COMMENT: (L62) Problems or instances?

R. FIXED, the correct term was DTLZ Problem’s instances.

1.4.7 COMMENT: (L77) What does the bias imply?

.

R. It was a typographical error. The intended meaning was intransitivity; hence, the sentence was corrected and appropriately complemented with a description of its implication and references. The final result in the manuscript was:

“In the case of outranking relationship, indicators of dominance or preference are defined given some thresholds. This approach’s main criticism is the difficulty obtaining the model parameters [8]; however, there are methods to solve it [12]. On the other hand, MAUT does not work when intransitivity exists between the preferential model [11]. The intransitivity phenomenon occurs in many real cases when exist a looping between the alternatives to select. It is important to consider this property to avoid possible incoherent solutions [Collete, 2004].”

Collette, Y., & Siarry, P. (2004). Multiobjective optimization: principles and case studies. Springer Science & Business Media.

1.4.8 COMMENT: (L91) Rewrite

R. We rewrite the corresponding paragraph.

“In Cruz et al. [8], the multicriteria ordinal classification requires the DM to separate solutions into two categories. In a preference incorporation method with this classifier, the human categorization is the stage with the lowest cognitive demand of the entire process.”

1.4.9 COMMENT: (L96) Rewrite

R. We rewrite the corresponding paragraph. The result was:

“Using outranking relationships allows handling the characteristics of many DMs facing real-world problems [8]. Being good that used preference incorporation methods meet the desirable characteristics described above. They are related to interaction with the DM, compatibility between the preferential model and the DM, properties of the preferences, and parameters' inference.”

1.4.10 COMMENT: (L134) It is unclear the meaning of this sentence. Please emphasize or delete.

R. We delete the sentence because it comments about the experimental results, and it seems disconnected from the rest of the paragraph related to the main contributions. This is the deleted sentence “Various versions of the proposed algorithm are analyzed to evaluate the preferences’ selective pressure and their impact on the algorithm performance”.

1.4.11 COMMENT: (L199) The techniques of NSGA-III came from the fronts? The selection is not an operator… or, is it? Please rewrite

R. A selection is indeed a genetic operator; however, NSGA-III already performs a careful elitist selection of solutions with its characteristic niche preservation operation; also, the use of reference points attempt to maintain an adequate diversity in the population during the evolutive process. Given the past two conditions, the selection is reduced in the framework to a random selection which in combination with the chosen mutation and crossover operators, produces the new offsprings. We rewrite the corresponding paragraph to respond the two questions. The result in the manuscript was as follows:

“The Nondominated Sorting Genetic Algorithm III proposed in [9] is a genetic algorithm similar to the original NSGA-II. They search the Pareto optimal set performing a non-dominated sorting. The difference is the maintenance of diversity with the selection operator. The first uses crowding distances, and the second uses reference points. NSGA-III discriminates between the non-dominated solutions using a utility function, which calculates a solution's relevance to approximate a reference point.”

1.4.12 COMMENT: (L206) Rewrite

R. We rewrite the corresponding paragraph. The resulting paragraph is:

“To incorporate a DM’s preferences, we propose integrating the ordinal classification method INTERCLASS-nC into the NSGA-III. The original work [8] only defines the classes "satisfactory" (Sat) and "unsatisfactory" (Dis); the DM gives a reference set to generate these classes (with one or more representative solutions for each class). This classification complements the non-dominated sorting to increase the capacity to discriminate solutions; this strategy induces a greater selective pressure, focusing the search toward the ROI. In this work, two classes are added internally for giving more precision in the comparison of the solutions:”

1.4.13 COMMENT: (L215) Rewrite to discard “said”

R. We rewrite the corresponding sentence as follow: “Let Qt the children population of the current generation with equal number of individual N of Pt .”

1.4.14 COMMENT: (L262) I am not sure about the use of the wedge

R. After consulting math bibliography, we decided to change the symbol wedge by comma.

1.4.15 COMMENT: (L268) Avoid the use of “said” for something that was established before.

R. We rewrite the paragraph to discard "said" and clarify the definition of the three used, the result in the manuscript was as follows:

“Euclidean distances: Minimum, mean, and maximum Euclidean distance among the obtained non dominated solutions and the ROI (also called Min Euclid, Mean Euclid, Max Euclid)”

1.4.16 COMMENT: (L291) Rewrite.

R. We rewrite the corresponding paragraph as follows:

“Table 4 shows the reached performance for each algorithm when solving each DTLZ problem. For space reasons, these results are only presented for two performance measures. The first two columns show the result for the original NSGA-III algorithm. The next columns present eight variants of NSGA-III with preferences. The first six columns correspond to variants without activating the solutions restarting strategy. The last two columns correspond to variants that use restarting to reduce the effect of incorporate preferences.”

1.4.17 COMMENT: (L299) It seems like the authors are going to detail a particular finding “if the DM is interested in solutions…” However, the finding is too vague. Please rephrase.

R. We rewrite the corresponding paragraph as follows:

“In this paper, the main measure to evaluate algorithms is related to the counting of highly satisfactory solutions because preferences elicitation is aligned with this measure. But considering other DM could be interested in the closeness to the ROI, the Euclidean distance is an alternative because it is frequently used in decision making. For a DM interested in highly satisfactory solutions, the best variant for all DTLZ problems is C100R0. In contrast, if the DM is interested in solutions closer to the ROI, we cannot find a unique variant as the best; They are dependent on the problem. The C100R0 variant offers solutions close to the ROI in four of the seven problems evaluated (DTLZ2 - DTLZ4, DTLZ7); For the DTLZ5 and DTLZ6 problems, C1R2 has a better performance. The original NSGA-III algorithm offers solutions closer to the ROI for the DTLZ1 problem. It is noteworthy. that C100R0 is never the worst option; the other variants are the worst at least once.”

1.4.18 COMMENT: (L304) Does it mean that every other option is at some point the worst choice?

R. That is correct. For example, in Table 5, the performance of C1R2 for the DTLZ6 problem is the best for the minimum distance measure. In contrast, C1R2 performs poorly in providing solutions closer to the ROI for the DTLZ1 problem. The final sentence of the previous response resumes this comment.

1.4.19 COMMENT: (L344) Rewrite.

R. We rewrite the corresponding paragraph as follows:

“This article presents a novel method for incorporating DM’s preferences into the NSGA-III algorithm, named NSGA-III-P. INTERCLASS-NC is a multi-criteria and outranking ordinal classifier that allows incorporating preference, giving the algorithm the capacity to improve the discrimination of solutions and intensify the search toward the region of interest. Excessive intensification can diminish the algorithm’s effectiveness. To regulate this selective pressure, we add two complementary strategies to the search in NSGA-III-P: control the activations of the classification and control the restarts of solutions.”

1.4.20 COMMENT: (L383) Rewrite.

R. We rewrite the corresponding paragraph as follows:

“Based on computational experimentation, the best alternative to the DTLZ problems is the C100R0 (always classify without restarts) when the DM is looking for highly satisfactory solutions. When the DM prefers solutions closer to the ROI, the variant C1R2 (classify and sometimes restart) and C100R0 have the best performance with two and four problems, respectively. In general, the proposed method NSGA-III-P outperforms NSGA-III because it allows obtaining better approximations to the ROI using the principal performance measures; only in one case, the NSGA-III is the best option for the DTLZ1 problem using the Max Euclidean distance.”

1.4.21 COMMENT: The Table 4 is too big. Perhaps it will be better to split it into two tables and optionally to reduce the font size.

R. Done, the font size was reduced.

Reviewer 2 Report

The article presents an ordinal multi-criteria classification method to integrate multiple decisions in a multi-objective evolutionary algorithm.  It is evaluated with the DTLZ standard problems for three objectives with different versions of the method proposed to the NSGA-III algorithm. The manuscript presents timely research on multiobjective evolutionary optimisation algorithms. The article is well-written and presented, however, a proofread is required to further improve the writing of the article by removing some errors such as. For example, define all the abbreviations for the first time of use. Also, avoid using the abbreviations in the abstract, i.e. NSGA. Remove all typographical/grammatical errors. The introduction can be further improved by adding more and relevant references and comparisons. 

Author Response

We are grateful for the reviewer's helpful comments and thank him for his careful reading of our work. We have considered these comments in the second version of our work. And we hope that we have met your concerns and expectations. Our responses to each of the reviewer's specific comments are listed below.

2.1 COMMENT: The article is well-written and presented, however, a proofread is required to further improve the writing of the article by removing some errors such as. For example, define all the abbreviations for the first time of use. Also, avoid using the abbreviations in the abstract, i.e. NSGA. Remove all typographical/grammatical errors.

R.           FIXED, the manuscript was carefully revised using an automated writing assistance software, and was proofread by an expert on English language.

2.2 COMMENT: The introduction can be further improved by adding more and relevant references and comparisons. 

R.           DONE.